# Experimental Benchmarking of Redox Flow Cells

**Adam H. Whitehead \***, **Alasdair Robertson**, **Benjamin Martin, Elisha Martin and Emma Wilson**

Invinity Energy Systems, Unit 1, 10 Easter Inch Road, East Inch Ind. Estate, Bathgate EH48 2FG, UK
* Correspondence: awhitehead@invinity.com

**Abstract:** There are increasing numbers of scientific articles dedicated to developments in the field of redox flow batteries. To date it is most common to provide efficiency values as a measure of performance. However, there are no agreed standard experimental conditions for these measurements, and so their merit as a tool for comparing different innovations among research groups is put into question. In the following manuscript, various experimental precautions are outlined to reduce experimental artefacts. Original experimental measurements on vanadium flow cells, together with data from the literature, are examined to explore efficiencies and two alternative benchmarking metrics: resistivity and self-discharge current density. The sensitivity of these parameters to current density, temperature, flow rate and state-of-charge range are examined, from which it is concluded that resistivity and self-discharge current density exhibit superior properties to efficiencies for quantifying flow battery improvements.

**Keywords:** vanadium redox flow battery; efficiency; performance benchmarking; resistivity; self-discharge; experimental precautions; current density; standard experimental condition

## 1. Introduction

A wide variety of flow battery chemistries and cell designs are under development [1,2]. Increasing reliance on intermittent power sources, such as wind, photovoltaic and tidal generators, has created a demand for large-scale energy storage systems. Redox flow batteries can easily provide many hours of energy storage and boast very long lifetimes [3] with low environmental impact [4–6] and favorable economics for high duty cycle application, [7,8] which makes them well-suited to meet this pressing need.

To date, vanadium is the most widely adopted chemistry, especially in commercial systems and will be used to exemplify various concepts hereafter. However, it should be understood that the principles can be more widely applied to study the redox flow cells of various chemistries.

Efficiency is a widely used measure of performance. Typically, energy efficiency, $\eta_E$, voltage efficiency, $\eta_V$, and coulombic efficiency, $\eta_Q$, are reported [9]. These terms are defined by common convention (e.g., ref. [10]) as:

$$\eta_E = \left| \frac{\int V \cdot j_{discharge} dt}{\int V \cdot j_{charge} dt} \right| \tag{1}$$

$$\eta_Q = \left| \frac{\int j_{discharge} dt}{\int j_{charge} dt} \right| \tag{2}$$

$$\eta_V = \frac{\eta_E}{\eta_Q} \tag{3}$$

where $j_{discharge}$ and $j_{charge}$ are cell current densities on discharge and charge, respectively; V is cell voltage; and the integrals are measured over the entire discharging and charging periods, respectively. It is important that the cycle starts and returns the electrolyte to the

same state-of-charge (SOC). The concept of SOC applying to the electrolyte, rather than the whole battery, highlights the division of energy and power (electrolyte and stack) that is found in flow batteries in contrast to conventional batteries. Additionally, current density is equal to cell current divided by the active area (typically the wetted, geometric membrane area).

From Equations (1)–(3) it is clear that only two efficiency terms are independent, the third being derived from the other two. Therefore, coulombic and voltage efficiencies will be discussed further, without more mention of energy efficiency.

Typically, when characterizing flow cells, cycling is performed at a constant current density, which is equal in magnitude for charging and discharging. In this case the above equations simplify to:

$$\eta_Q = \frac{t_{discharge}}{t_{charge}} \tag{4}$$

$$\eta_V = \frac{\int_{discharge} V dt}{\int_{charge} V dt} \cdot \frac{t_{charge}}{t_{discharge}} \tag{5}$$

where $t_{charge}$ and $t_{discharge}$ are the times taken for charging and discharging, respectively, for a given cycle.

It may also be noted that under any given conditions the cell under test will exhibit internal resistivity, $\rho$, and self-discharge current density, $j_{sd}$. The first term is self-explanatory, the second term may be considered as the difference in current density stored chemically in the electrolyte to that flowing through the current collectors of the cell. For example, if a given current density is applied during charging, the electrolyte will increase in SOC by less than the expected amount based on the concentration of electroactive species. This difference is due largely to permeation of active species through the membrane, causing internal discharge reactions [11]. With this in mind, $j_{sd}$ may be seen to relate to $\eta_Q$ and $\rho$ to $\eta_V$. The relative merits of using these less-common metrics will be explored.

It Is worth pointing out at this stage that $\rho$ for the cell can be directly compared to values of through-plane resistivity (sometime referred to as area specific resistance) of bipolar plates, membranes and electrodes, being expressed in the same units. This may instantly provide some guidance as to the relative importance of the individual cell component contributions to the overall resistivity.

This work expands on a topic brought up at a recent conference [12].

## 2. Materials and Methods

### 2.1. Materials

Test cells were required to exemplify the metrics being investigated. The materials in these cells are therefore only of secondary importance, because it is the analytical method that is being explored. Nonetheless, the test cells were of $5 \times 5$ cm$^2$ active area with simple flow-through graphite felt electrodes, separated by a commercial ion-exchange membrane. The electrodes were purchased in an "activated" state. The electrolytes were commercial Generation 1 types (i.e., vanadium sulfates in sulfuric acid) [3,13].

Reference cells were built in a similar manner to test cells, comprising similar materials but with smaller active areas, typically ~2 cm$^2$.

### 2.2. Measurement Set-Up

Hydraulically, the reference cell was arranged in line with the test cell, Figure 1. Although a parallel configuration could have been used, this arrangement ensures the entire flow passes through the test cell, simplifying flow rate determination. The reference cell voltage was measured under open-circuit conditions. The test cell voltage was measured, and external current applied, with a commercial high precision battery cycler or purpose-built test equipment.

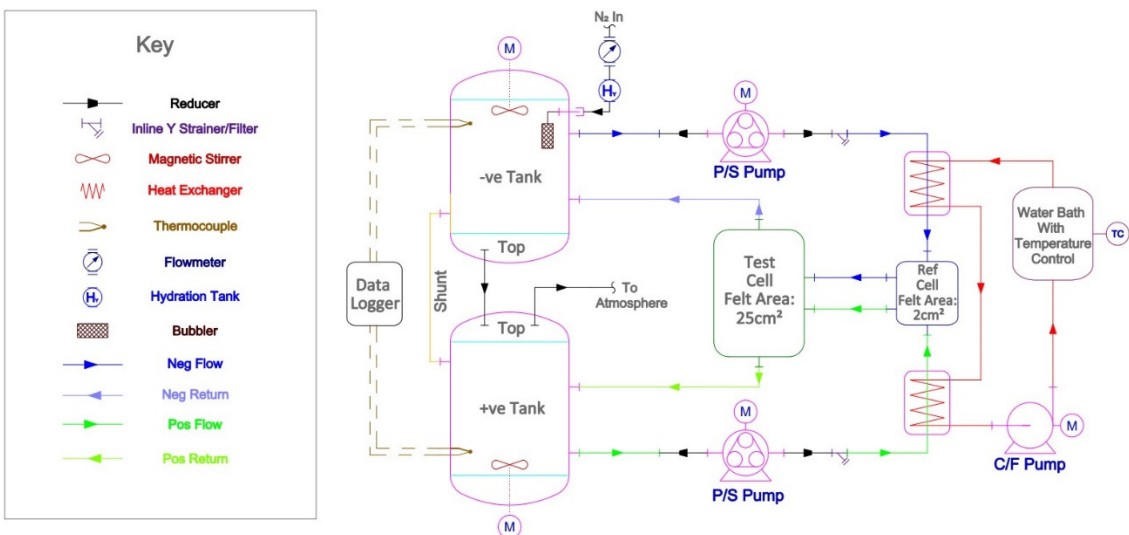

**Figure 1.** Schematic representation of the test-rig used in this study. Note that the test cell voltage, reference cell voltage and thermocouple outputs were recorded by the battery cycler, which additionally supplied current to the test cell. "Shunt" in this diagram refers to the hydraulic shunt between tanks.

The electrolytes were pumped in a circuitous fashion from tanks to reference cell/test cell and back, using peristaltic pumps. To protect against aerial oxidation, humidified nitrogen was gently sparged through the negative electrolyte, before flowing through the headspace of positive tank. Temperature was measured using T-type, hermetically sealed thermocouples immersed directly in the electrolyte within the reservoirs. The electrolyte temperature was controlled by means of a circulating water heater/chiller. Deionized water was employed as the heat exchange fluid, which was pumped through a purpose-built, in-line, plastic heat exchanger.

The SOC of the electrolytes was determined from the reference cell voltage using an empirical equation to account for complexation of the vanadium species and non-unity value of the activity coefficients [14]. It was assumed that the temperature of electrolyte in the reference cell was equal to that in the tanks, which is likely, given the rather short transit times between tanks and reference cell.

The positive and negative electrolyte tanks were connected by a long, thin, open flexible hose (hydraulic shunt) to passively maintain equal electrolyte volumes, without the excessive crossover of electrolytes. This had previously been shown to be beneficial in rapidly reaching a stable discharge capacity [15,16].

The vanadium half-cell reactions are:

$$\text{Negative electrode reaction}: \text{V}^{3+} + \text{e}^- \leftrightarrow \text{V}^{2+}$$

$$\text{Positive electrode reaction}: \text{VO}_2^+ + 2\text{H}^+ + \text{e}^- \leftrightarrow \text{VO}^{2+} + \text{H}_2\text{O}$$

## 3. Results and Discussion

### 3.1. Resistivity, $\rho$, and Voltage Efficiency, $\eta_V$

#### 3.1.1. General Considerations

On the application of current density through a cell, the cell voltage will change by an amount, $j \cdot \rho$ (where $j$ is defined throughout as positive during charging and negative for discharging), from the open-circuit voltage (OCV). Although this statement appears self-evident, it hides a couple of implicit assumptions, namely:

1.  current density is constant over the whole surface of the membrane in the test cell;
2.  current density through the membrane is equal to current density through the electrodes.

The first of these is clearly an approximation, because the SOC of the electrolyte (and hence the OCV) is changing in the direction of flow through the test cell. The current density is highest in the inlet region and lowest at the outlet. This is problematic because the reference cell measures the OCV of electrolyte at the inlet of the test cell, $V_{ref}$. The standard approach of comparing cell voltage to reference cell voltage (Equation (6)) will incorrectly overcalculate the resistivity because of the local difference in current density.

$$\rho = \frac{V - V_{ref}}{j} \tag{6}$$

The error in $\rho$ may be considered in terms of the "missing" energy as $j \cdot (j_{inlet} - j) \cdot \rho_{true}$, where, as before, $j$ is the average current density, $j_{inlet}$ is the local current density at the test cell inlet and $\rho_{true}$ is the genuine cell resistivity. The origin of this "lost" energy is clearly in the formation of a gradient in SOC through the cell. This gradient is removed in the tank, on the assumption that mixing in the tank is rapid. Therefore, the "lost" energy in the cell actually appears as the free energy of the mixing of the electrolyte exiting the test cell, with electrolyte at a different SOC in the tank. Therefore, it should be apparent that the calculated resistivity from Equation (6) approaches the true value when the change of SOC through the cell, $\Delta$SOC, tends to zero

To the second point listed above, it is clear that all of the current flowing through the current collectors is involved in charge-transfer reactions within the test cell. However, as soon as current is carried ionically, a part will bypass the test cell membrane and pass through the electrolyte lines and through the membrane in the reference cell. However, a quick calculation shows that the resistance of the bypass through the reference cell is generally sufficient that <0.1% of the current flows through the reference cell (e.g., 25 cm$^2$ of membrane in the test cell has a resistance of ~0.02 $\Omega$, compared to 10 cm of fluid line with 5 mm internal diameter and 3 $\Omega$ cm electrolyte resistivity, which has a resistance of ~150 $\Omega$). Therefore, generally when testing small single flow cells this term can be ignored. However, it may become significant if this approach is employed for stacks of cells or at very high current densities. In these latter cases, the main error is in determining $V_{ref}$ and hence the SOC.

It should be noted that both points above will affect voltage efficiency in a similar manner to resistivity, causing similar errors.

### 3.1.2. Flow Rate Dependence

Qiu et al. [17] and Chen et al. [18] have shown theoretically that at low flow rates, resistivity is impacted by vanadium species diffusion to the electrode fibers, but as linear flow velocities increase to relatively modest values, convection through the cell (i.e., SOC change from inlet to outlet, as alluded to in Section 3.1.1) becomes the dominant influence. Goulet et al. [19], using micro fluidic experiments on carbon papers, demonstrated that the negative electrode exhibited a higher overpotential than the positive at a given $j$ and was flow-independent at least down to 0.2 cm s$^{-1}$, and up to 100 mV overpotential. However, Roznyatovskaya et al. [20] have pointed out the very wide range of reported rate constants for vanadium redox reactions on graphite electrodes and surmised that "it is difficult to state which half-cell reaction would limit the VRFB operation. However, the anodic reaction is suggested to limit the performance of a VRFB from kinetics aspects even at low current densities, thereby the cathodic reaction appears to exhibit mass transport effect at high current densities."

There have also been concerns raised about flow homogeneity and potential disruption of fiber–fiber interconnections at particularly low and high flow rates, respectively [21–23].

In any case, $\rho$ and $\eta_V$ would be expected to demonstrate significant flow dependence, tending to limit values at higher flow rates. Therefore, the flow rate and flow velocity must be specified, and in many cases relatively high flow rates should be used to minimize the contributions from convection and diffusion terms.

### 3.1.3. Dependency on SOC

A test cell was cycled under constant current conditions at 100 mA cm$^{-2}$ with vanadium electrolyte thermostatically controlled at 24.4 ± 0.3 °C over the whole cycle (A = 25 cm$^2$, flow rate = 128 cm$^3$ min$^{-1}$ per half-cell). Figure 2 shows the resistivity for a typical cycle, arranged against SOC of electrolyte entering the test cell. Despite the relatively small change in SOC through the cell (ΔSOC = 0.007), it can be seen that there was some increase in resistivity at high and low SOC. Therefore, the voltage efficiency is dependent on the range of SOC covered in a test cycle. Short cycles around SOC = 0.50 would exhibit higher $\eta_V$ than cycles covering a wider SOC range.

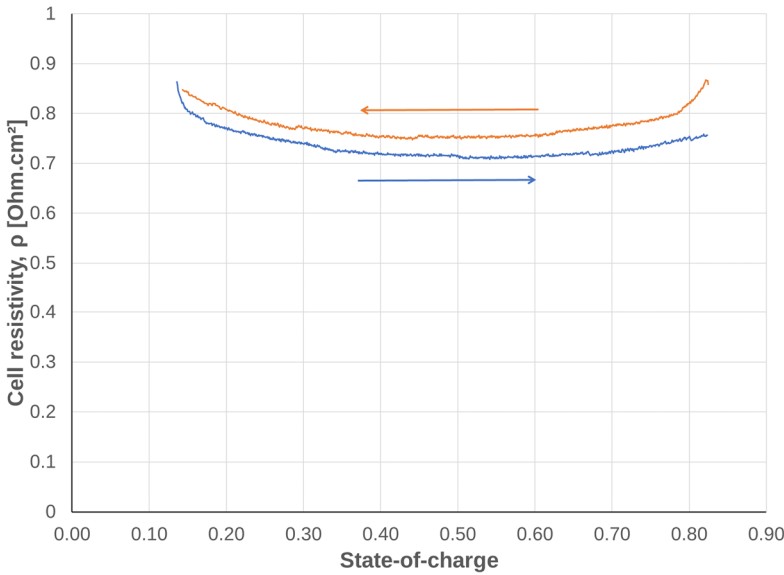

**Figure 2.** Resistivity of a vanadium flow cell, during charge (blue curve) and discharge (orange) arranged against electrolyte inlet state-of-charge, at 24.4 ± 0.3 °C electrolyte temperature.

In order to provide a single value for resistivity for facile comparison, it is proposed to take the average resistivity during charge and discharge at SOC = 0.50 at a given temperature. At around SOC = 0.50 the resistivity is almost independent of SOC and so small errors in its determination are likely to be irrelevant. This value will be referred to as the 50% resistivity, $\rho_{50\%}$. Another reason for selecting 50% is that, at this SOC, there is unlikely to be much impact from the, so called "power drop effect" on discharge when employing anion–exchange membranes [24–26].

### 3.1.4. Dependency on Temperature

To investigate the temperature dependence of $\rho_{50\%}$ and $\eta_V$, a cell was cycled repeatedly over a range of electrolyte temperatures from 16 °C to 41 °C (A = 25 cm$^2$, flow rate = 128 cm$^3$ min$^{-1}$).

Figure 3 reveals the significant temperature sensitivity of the performance of a vanadium flow cell. Similar sensitivity has been observed for a wide range of flow-through electrode materials, ion-exchange membranes and electrolyte compositions. It is clear that the direct measurements of the electrolyte temperature are required in order to obtain accurate $\rho_{50\%}$ or $\eta_V$ values. In many cases only ambient air temperature is reported, although this may deviate from electrolyte temperature by several degrees, especially at high current densities. By way of example, from Figure 3, an offset in temperature from 30 °C by 1 °C, would cause a change in $\rho_{50\%}$ of 0.027 Ω cm$^2$ ≡ 2.4%.

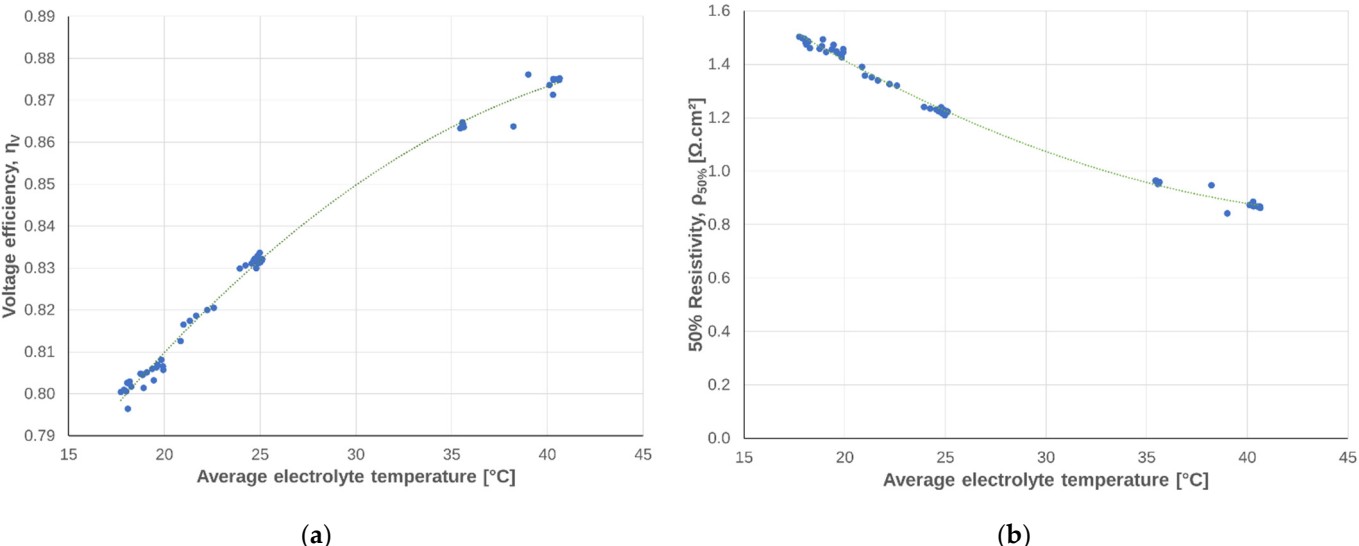

**Figure 3.** A vanadium redox flow cell was cycled under constant current conditions, over a range of electrolyte temperatures, from which: (**a**) voltage efficiency and (**b**) 50% resistivity were calculated for each cycle.

For the sake of standardization, a temperature of 25 °C or 30 °C for vanadium chemistry would seem appropriate, while for other chemistries different temperatures may be selected to better reflect the average operating temperatures. If the correlation between $\eta_V$ and temperature is known, then the experimental data may be mathematically corrected to a standard temperature.

### 3.1.5. Dependency on Current Density

A vanadium flow cell was cycled three times at current densities of 40, 60, 80 and 100 mA cm$^{-2}$. The average $\rho_{50\%}$ values after temperature correction were all found to lie within $1.049 \pm 0.008$ $\Omega$ cm$^2$, with no obvious trend on current density. Although this range of current densities is quite modest, and at around 50% SOC mass transport limitations would not be expected, it has often been observed that vanadium flow cells show ohmic response over a wider range of current densities in polarization experiments [22,27–29].

In contrast, if resistivity is independent of current to a first approximation and also relatively constant over the range of cycling, as exemplified in Figure 2, then it would be expected that $\eta_V$ would be very dependent on current, as is invariably found [30,31]. Because there is no universally agreed single cell testing current density, $\rho_{50\%}$ presents a more convenient means to compare tests than voltage efficiency.

### 3.2. Self-Discharge Current Density, $j_{sd}$, and Coulombic Efficiency, $\eta_Q$

#### 3.2.1. General Considerations

Electroactive species crossing the ion exchange membrane within a flow cell lead to discharge reactions [32], with at least some of the discharge occurring within the membranes [33]. Elgammal et al. have suggested expressing the flux of electroactive species as a discharge current density [34].

It is also known that the relative importance of transport through diffusion, convection (i.e., electro-osmotic convection and osmosis) and migration are dependent on current density, SOC and temperature. Therefore, at first glance, it would seem that $j_{sd}$ is not going to be a more useful metric than $\eta_Q$ for comparing flow cells.

It may also be noted that under normal operating conditions for vanadium chemistry, $j_{sd}$ and $\eta_Q$, result from membrane transport exclusively. However, excessively high voltages or the presence of catalytic impurities (or deliberate additions [35]) can accelerate parasitic

reactions (e.g., $H_2$ formation) to such a degree that they also impact the values of these metrics.

Membrane transport processes will occur not just through the test cell, but also through the reference cell. Therefore, it is advised to use the same membrane in the reference and test cells and to minimize the reference cell membrane area in comparison to that of the test cell as far as possible.

### 3.2.2. Dependency on SOC

Despite the reservations expressed above, if it is assumed that for a given cell at a given electrolyte temperature and current density $j_{sd}$ is independent of the SOC, then it follows that the useful charge stored chemically during charging is smaller than the applied electrical charge. In other words, the useful current density stored chemically in the cell is less than the applied current density on charging by $j_{sd}$, and equally that the external discharge current density is smaller than the effective discharge of the chemical store by $j_{sd}$. Therefore, the SOC, which is a measure of the charge usefully stored in the electrolyte, would be dependent on the integral of $(j - j_{sd})$ over time. To test this assumption two vanadium cells were repeatedly cycled at a constant current density. Figure 4a shows that in this example, with a cation exchange membrane, a fixed $j_{sd}$ of $4.0 \, \text{mA cm}^{-2}$ was sufficient to remove the hysteresis between charge and discharge on repeated cycles. The results with an anion exchange membrane are shown in Figure 4b. In this case, more cycles were used and the electrolyte was $5\,^\circ\text{C}$ warmer than for the cation exchange membrane, because the hysteresis per cycle was rather small. In this case, $j_{sd} \sim 0.5 \, \text{mA cm}^{-2}$. Anion exchange membranes have typically been observed to have lower vanadium permeability and higher coulombic efficiency under given test conditions than cation exchange membranes [36–38]. In both of these examples, a $j_{sd}$ term that was independent of SOC appears, at a first approximation, reasonable.

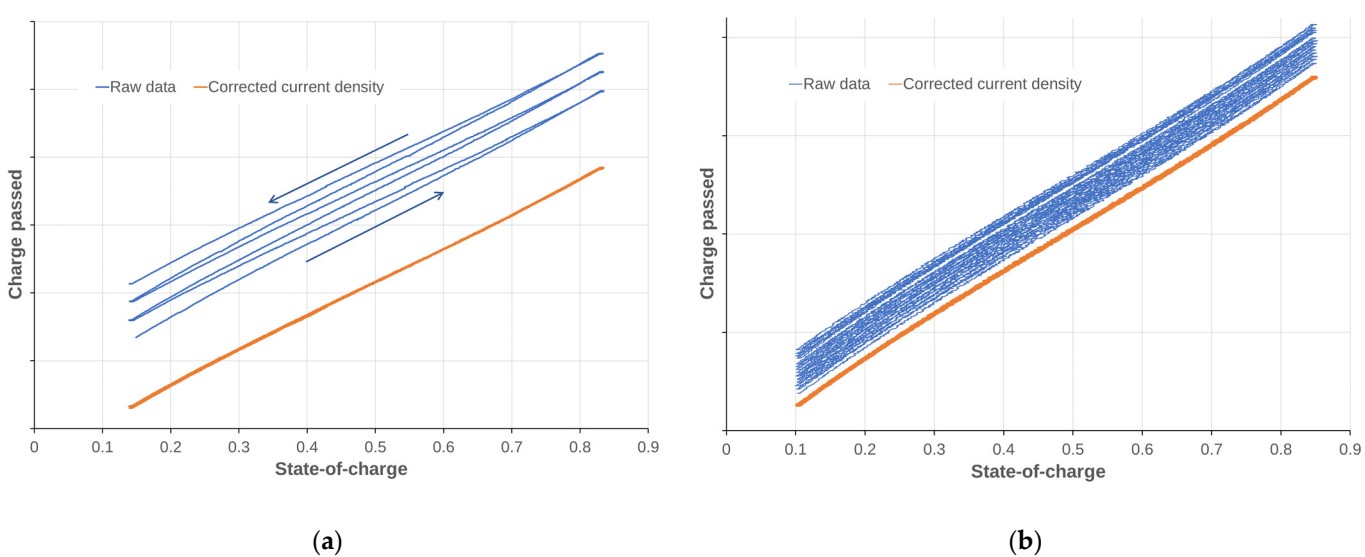

**(a)**            **(b)**

**Figure 4.** Repeated cycles of a vanadium redox flow cell at $100 \, \text{mA cm}^{-2}$ showing the cumulative electrical charge passed (raw data) and the charge passed after correcting j by a constant $j_{sd}$ term (corrected current density) arranged against the measured SOC of the electrolyte, for (**a**) cycles 4 to 6 of a cell with cation exchange membrane at $25\,^\circ\text{C}$ and (**b**) cycles 4 to 17 of a cell with anion exchange membrane at $30\,^\circ\text{C}$.

The observation of SOC independence is unexpected but, to the authors' knowledge, universal for ion exchange membranes tested in commercial-strength vanadium electrolyte at these current densities and temperatures. It is most likely because the difference in $j_{sd}$

at high and low SOC is relatively small and the method employed here gives an average value over the cycle.

This allows a very useful relationship to be drawn regarding flow cells cycled under constant current conditions at constant temperature, namely that:

$$j_{sd} = \frac{j(1 - \eta_Q)}{1 + \eta_Q} \qquad (7)$$

In contrast to $j_{sd}$, which may be taken over partial or multiple cycles, with no requirement to start and end at the same SOC, $\eta_Q$ can be determined only for cycles that start and end at exactly the same SOC. A mismatch between start and end SOC of 1%, would be sufficient to cause an error in $\eta_Q \geq 0.01$. This may be very significant, considering that $\eta_Q$ is typically $> 0.95$ in reported data. The use of cut-off voltage limits, rather than SOC limits, may also lead to misleading values [39].

It should also be noted that $j_{sd}$ is rather small compared to $j$; therefore, care must be taken to ensure that measuring equipment is suitably calibrated for this purpose. This is equally true for $\eta_Q$ measurements.

### 3.2.3. Dependency on Temperature

A redox flow cell was repeatedly cycled over a range of electrolyte temperatures. Figure 5 shows that although there is significant experimental scatter, there was also a clear trend to increasing $j_{sd}$ with temperature. This is probably not surprising as a similar decrease in $\eta_Q$ with increasing temperature would also be expected and has been reported for cells containing various membranes [40,41].

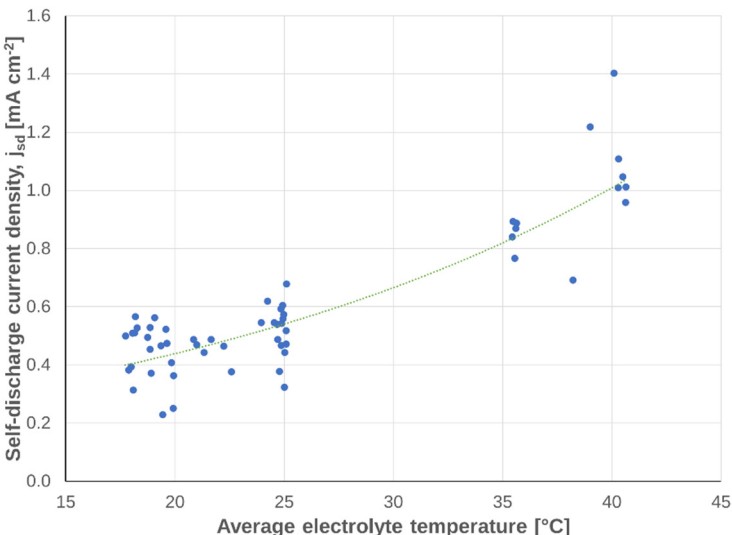

**Figure 5.** Self-discharge current density of a vanadium redox flow cell under constant current cycling conditions of 100 mA cm$^{-2}$ over a range of electrolyte temperatures. Each point represents the calculated $j_{sd}$ of a single cycle. The regression line is intended as a guide to the eye only.

If $j_{sd}$ is used as a metric for comparison, it is advisable to take an average over multiple cycles at a given temperature to reduce experimental scatter, as exemplified in Figure 5.

### 3.2.4. Dependency on Current Density

Gandomi et al. found a very strong dependence on $j_{sd}$ (which they called crossover current density) on overpotential, which would imply a strong dependence on $j$ [42]. This is probably expected, when one considers that migration and osmosis will become more important than diffusion at higher current densities. Nonetheless, $\eta_Q$ almost always increases at higher $j$, regardless of the type of ion-exchange membrane [43,44]. One exception arises

if the test cell voltage is allowed to go too high with increasing j, such that side reactions cause a notable decrease in efficiency.

To investigate the dependence of $j_{sd}$ and $\eta_Q$ on j experimentally, a cell was subjected to repeated cycles, whereby 2 to 4 cycles were repeated at each current density and average $j_{sd}$ values determined by the method illustrated in Figure 4. This also included charging the cell to SOC = 0.85 and measuring the rate of change of SOC with time under open-circuit conditions (rate of spontaneous cell discharge), without interrupting the electrolyte pumping, to obtain a value of $j_{sd}$ at j = 0 mA cm$^{-2}$. These data are shown in Figure 6.

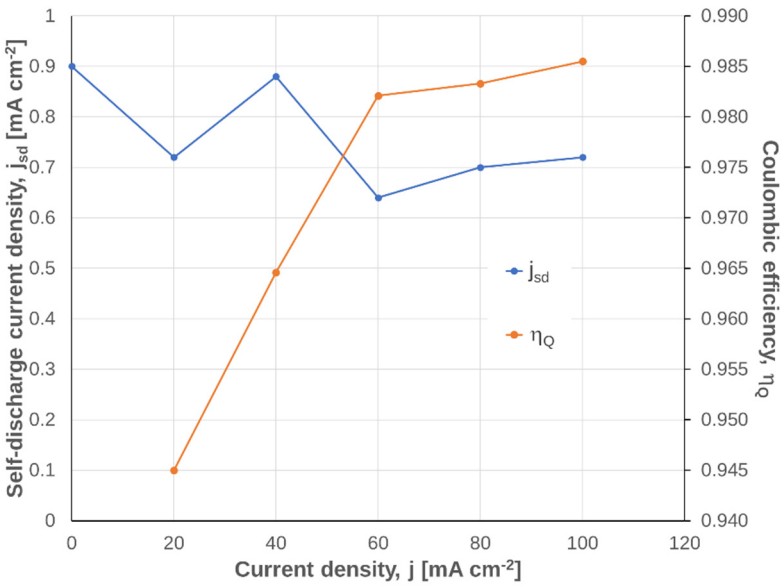

**Figure 6.** Self-discharge current density and coulombic efficiency arranged against current density for a vanadium flow cell, cycled under constant current conditions at 31.6 °C.

As expected, $\eta_Q$ (determined using the ratio of discharge to charge times, Equation (4)) increased monotonically with j. There appears to be a slight decrease in $j_{sd}$ with increasing j for the membrane employed, although an average $j_{sd} \sim 0.7$ mA cm$^{-2}$ over j = 20–100 mA cm$^{-2}$ could be concluded. This is unexpected and may only relate to the specific membrane chosen in this study; therefore, a wider exploration of literature values was undertaken using the relationship given in Equation (7). Unfortunately, most publications give a value of $\eta_Q$ at only a single current density and those that cover a range of j tend to show the efficiencies graphically. Nonetheless, numerical values were given in several publications, although the electrolyte temperature was not specified in any of them. $J_{sd}$ values were derived from the given coulombic efficiencies and are shown in Figure 7.

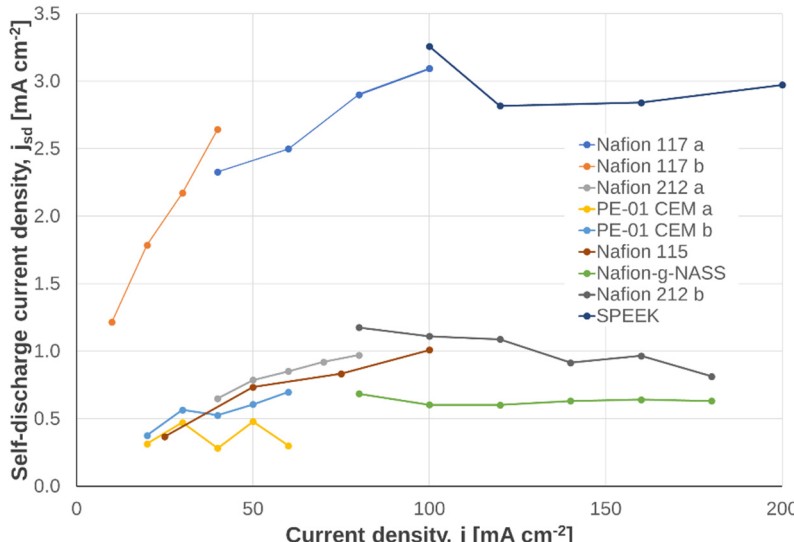

**Figure 7.** Self-discharge current densities of vanadium flow cells with a range of membrane types, derived from the literature $\eta_Q$ values under constant current cycling conditions. Note that electrolyte compositions varied and electrolyte temperatures were not given. The information sources were: Nafion 117 a [45], Nafion 117 b [46], Nafion 212 a [47], PE-01 CEM a [48], PE-01 CEM b [49], Nafion 115 [50], Nafion 212 b and Nafion-g-NASS [51], SPEEK [52].

It should be emphasized that the experimental conditions used to obtain the data shown in Figure 7 were not fully described and most likely differed between different sources. For example, it is understood that electrolyte composition affects vanadium species transfer rates through the membrane [53]. Therefore, the derived $j_{sd}$ values in Figure 7 cannot be used directly to compare different types of membranes. However, what is important to notice is that some membranes exhibited relatively little dependence on j, whereas the Nafion membranes especially exhibited increasing $j_{sd}$ with j. Hence, $j_{sd}$ appears more or less dependent on j, depending on the membrane type and experimental conditions. However, unlike $\eta_Q$, $j_{sd}$ measured at several different values of j provides a clearer picture of the dependence of current-dependent electroactive species transport (migration and electro-osmotic convection).

### 3.2.5. Additional Information

A hydraulic shunt is a useful means to reach a pseudo-equilibrium state more rapidly, in terms of electrolyte composition during cycling experiments. However, it also gives a higher value of $j_{sd}$ than experiments without the shunt (in which the volumes of electrolyte are allowed to differ freely). This is because electrolyte is gradually transferred through the shunt into the opposing tank, causing a chemical discharge reaction. The optimum design of the hydraulic shunt can minimize but not eliminate this effect.

Self-discharge current density may also be used when studying flow battery stack performance. However, in this case, there will also be a contribution from the shunt currents [54], which will lead to higher $j_{sd}$ (lower $\eta_Q$) than for a single cell.

### 4. Conclusions

Table 1 is a summary of the sensitivity of the various metrics, as discussed above. As can be seen, $\rho_{50\%}$ and $j_{sd}$ exhibit less sensitivity to typical experimental variables than voltage and coulombic efficiencies.

**Table 1.** Comparison of the sensitivity of the benchmarking metrics outlined above.

| Metric | Sensitivity | | |
|---|---|---|---|
| | Cycle Limits (SOC Limits) | Current Density | Electrolyte Temperature |
| Voltage efficiency, $\eta_V$ | Some | Strong | Strong |
| 50% Resistivity, $\rho_{50\%}$ | None | Almost none | Strong |
| Coulombic efficiency, $\eta_Q$ | Strong | Strong | Strong |
| Self-discharge current density, $j_{sd}$ | Almost none | Almost none to strong | Strong |

Plotting $j_{sd}$ against j also gives a graphical impression of the dependence of electroactive species cross-over in the current density region of interest in a way that is not simply apparent with $\eta_Q$.

$\rho_{50\%}$ is expressed in the same way as the area specific resistivity of various cell components, allowing relatively facile comparison (which is not possible with $\eta_V$).

Therefore, it is recommended that $j_{sd}$ and $\rho_{50\%}$ are more informative terms that should be widely adopted as metrics for describing redox flow cell performance and that efficiencies be better reserved for benchmarking full energy storage systems.

**Author Contributions:** Conceptualization, A.H.W.; formal analysis, A.H.W.; investigation, A.R., B.M., E.W. and E.M.; writing—original draft preparation, A.H.W.; writing—review and editing, A.R. and B.M. All authors have read and agreed to the published version of the manuscript.

**Funding:** This research received no external funding.

**Data Availability Statement:** Not applicable.

**Conflicts of Interest:** The authors declare no conflict of interest.

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
