# Peer review of "Experimental Benchmarking of Redox Flow Cells"

_batteries, doi:10.3390/batteries8110207_

Round 1

Reviewer 1 Report

The manuscript reported the various experimental precautions, to reduce experimental artifacts for VRFB tests. Original experimental measurements on VRFBs and data from the literature are examined to explore efficiencies, resistivity, and self-discharge current density. It can be concluded that resistivity and self-discharge current density exhibit superior properties to efficiencies in quantifying flow battery improvements.

I consider the content of this manuscript will definitely meet the reading interests of the readers of the Batteries journal. However, there are certain English spelling and grammar issues, and also the discussion and explanation should be further improved. I suggest giving a minor revision and the authors need to clarify some issues or supply some more experimental data to enrich the content. This could be comprehensive and meaningful work after revision.

1. For grammar issues, it is suggested that the author double-check the small grammar errors in the full text, especially the lack of and redundant use of definite articles. Moreover, references should not be cited in the abstract.

2. For the Keywords, experimental precautions, current density, and standard experimental condition should be added in order to attract a broader readership.

3. Line 24, ‘A wide variety of flow battery chemistries and cell designs are under development [2,3]. This is correct, but what is the main route of RFB? Why has it become a research hotspot? It should be explained better. For example, the implementation of renewable energy sources such as solar, wind and hydropower in the replacement of fossil fuels is progressively gaining momentum. However, most renewable energy sources are intermittent, opening spatial and temporal gaps between the availability of the energy and its consumption by the end-users. To address these issues, it is necessary to develop suitable large-scale energy storage systems for the power grid, such as RFBs [Electrochimica Acta 309 (2019): 311-325; 10.1016/j.renene.2022.05.129].

4. Line 25, vanadium is the most widely adopted chemistry, especially in commercial systems. Indeed, vanadium is not very cheap, if compared to more cost-effective zinc and iron-based RFBs [10.1088/1361-6463/ac4182; Journal of Power Sources 493 (2021): 229445]. What causes VRFB to be highly praised in the field of commercialization? It should be further clarified.

5. Line 51, This difference is due largely to permeation of active species through the membrane, causing internal discharge reactions [6]. With this in mind, jsd may be seen to relate to ηQ and ρ to ηV. If the difference is due to permeation, it is for sure related to CE (related to permeation). But why is the jsd also related to VE? VE is more related to the internal resistance of the VRFB, but not related to the permeation.

6. I think the statement 'SOC of the electrolyte' in the manuscript is worth discussing since the usual SOC is for the battery/VRFB itself.

7. Why the permeability of AEM is much lower than the CEM? The structure differences and the permeation mechanism differences should be explained better. For example, CEMs are mainly perfluorinated sulfonic acid matrices, and -SO3- groups are negatively charged. Hence, cations can be passed through CEMs easily. Since vanadium species are cations, the CEMs typically do not act as an efficient barrier layer toward vanadium species [10.1016/j.electacta.2021.138133].

8. In Figure 6, although various membranes are shown, but only Nafion-g-NASS is a composite/modified membrane. I suggest adding more results about other inorganic-organic hybrid membranes and polymer blending composite membranes. I think such a result will be more widely applicable.

Author Response

We thank the reviewer for their detailed and constructive suggestions.

1) We have reviewed the text and made minor changes, including removal of some redundant definite articles. Although in most cases we don't consider their original use to be incorrect, we hope that these changes have made the manuscript easier to read.

2) Thank you for the suggestions. These have been added.

3) A couple of lines have been added to the introduction to explain the current interest in flow batteries.

4) While we agree that the present dominance of vanadium over other flow battery chemistries has not been explained, our intent was to present methods for benchmarking all flow batteries. Vanadium was just used to exemplify the techniques and will be familiar to many researchers in the field. We consider a comparison between economic and technological aspects of different flow batteries may unnecessarily alienate some researchers, and lead them to see these as "vanadium only" metrics.

5) jsd is not related to VE. ". . . jsd may be seen to relate to ηQ and ρ to ηV." jsd is related to CE only. Resistivity is related to VE only.

6) An extra line has been inserted to cover this observation.

7) & 8) Both of these points are well taken. We fully agree that these are interesting questions. Our intent was to present techniques whereby researchers working on novel membranes could more readily explore and compare these issues. They are undoubtedly in a better position to provide answers.

Reviewer 2 Report

The report, titled "Experimental Benchmarking of Redox Flow Cells," demonstrates and optimizes the operating conditions of a redox flow battery. which is more suitable and appropriate to the journal's subject matter and a wide spectrum of readers. Though the manuscript draft was well organized, there are still a few issues that should be rectified. which are given below as a list.

·         A schematic diagram of the redox flow battery with the author's experimental setup and a brief description of its working principle is essentially needed.

·         Include a vanadium redox chemistry equation with a half-cell reaction.

Author Response

We thank the reviewer for their comments.
A schematic diagram of the flow battery has been added to the manuscript.
The vanadium half-cell reactions have also been added.

Reviewer 3 Report

Submitted paper represent a experimental evaluation in the field of battery cells considering a test program developed by authors due to lack of standardized test application. As also mentioned in specific literature the results are analyzed considering some parameters like voltage efficiency and resistivity in relation to electrolyte temperature, self discharge current density in relation to current density and cell resistivity considering the state of charge.

After carefully reading the submitted paper I consider that worth to be accepted in this form for publication and I encourage the authors to continue the good work in this field of research which is one of the key directions considering actual renewable energy application in various domains.

Author Response

We thank the reviewer for their very encouraging comments.